# **Tropospheric NO<sub>2</sub> concentrations over West Africa are influenced** by climate zone and soil moisture variability

Ajoke R. Onojeghuo<sup>1\*</sup>, Heiko Balzter<sup>1, 2</sup>, Paul S. Monks<sup>3</sup>

<sup>1</sup>University of Leicester, Centre for Climate Research, University Road, Leicester, LE1 7RH

<sup>2</sup>National Centre for Earth Observation, University of Leicester, University Road, Leicester, LE1 7RH

<sup>3</sup>University of Leicester, Department of Chemistry, University Road, Leicester, LE1 7RH

Correspondence to: Ajoke R. Onojeghuo (aroo1@le.ac.uk, a.onojeghuo@gmail.com)

10

5

#### Abstract.

The annual cycles of soil moisture and NO<sub>2</sub> have been analysed across the climate zones of West Africa using two satellite data sets (OMI on AURA and ASCAT on MetOp-A). Exploring the sources and sinks for NO<sub>2</sub> it is clear that the densely populated urban cities including Lagos and Abuja had the highest mean NO<sub>2</sub> concentrations (>1.8  $\times$  10<sup>15</sup> molecules cm<sup>-2</sup>)

- indicative of the anthropogenic urban emissions. The data analysis shows that rising soil moisture levels may influence the sink of NO<sub>2</sub> concentrations after the biomass burning. The results also show significant soil moisture changes in areas of high humidity especially in the east equatorial monsoon climate zone where most of the Niger delta is located (4 %/yr.). A decline in NO<sub>2</sub> (0.9 %/yr.) was also observed in this climate zone. Beyond seasonal linear regression models, climate based Granger's causality tests show that tropospheric NO<sub>2</sub> concentrations from soil emissions in the arid steppe (Sahel) and arid
- desert climate zones of West Africa are significantly affected by soil moisture variability (F > 10, p < 0.01). The arid steppe and arid deserts regions showed no significant changes in soil moisture levels but significant increase in tropospheric NO<sub>2</sub> concentrations (> 0.8 %/yr). The results demonstrate the critical sensitivity of the West African emissions of NO<sub>2</sub> on soil moisture and climate zone.

# 25 1 Introduction

Nitrogen oxides  $(NO_x)$  play critical roles in many atmospheric processes including the catalytic production of tropospheric ozone and the formation of nitric acid being key elements of local air quality with effects felt across global tropospheric chemistry (Richter et al., 2005). Atmospheric pollution affects human and ecosystem health as well as being intrinsically linked to climate in the present and coming decades (von Schneidemesser et al., 2015). NO<sub>2</sub> is a pollutant and key precursor

for tropospheric Ozone (O<sub>3</sub>) formation (Monks et al., 2015) and has both anthropogenic (fossil fuel / bio-fuel burning and incomplete combustion) and natural sources (soil emissions). West Africa is known to generate large amounts of pollutants from both its megalopolis concentrated around the Gulf of Guinea (Hopkins et al., 2009) and from biomass burning episodes during the Northern hemisphere dry season (Pradier et al., 2006) which occurs before the planting/rainfall season (Ker,

1995). Urban vehicular pollution is also rather high in urban West Africa owing to the high usage of often poorly maintained second hand cars and low quality fuel (Schwela, 2009). It has been estimated that in addition to seasonal biomass burning which releases  $NO_2$  and pyrogenic carbon into the atmosphere (Lehsten et al., 2009) directly, soil emissions of NOx represent 15% of global N emissions (Hudman et al., 2012).

5

10

Soil moisture conditions are an aggregated expression of the hydrological regime in the area which encompasses rainfall, evapotranspiration, surface runoff and groundwater supply (Ibrahim et al., 2015). Soil moisture trends affect the nitrogen (Keller and Reiners, 1994) energy, water and carbon exchange processes between the land surface and atmosphere (Basara and Crawford, 2002). It is also a determinant of the type and condition of vegetation in addition to overall ecosystem health in a region (GCOS, 2015). Over parts of West Africa, decadal-scale trends in rainfall have been analysed and showed flooding in the western Sahel between 2008 and 2010 (Hoscilo et al., 2015). Information on the temporal dynamics of soil moisture is important to identify the start of the wet season and drought events, especially in semi-arid/Sahel regions, where

- vegetation growth is driven by soil moisture variations (Ibrahim et al., 2015). IPCC (2007) projected that the effect of climate change on soil moisture will vary with soil characteristics, such that soils with lower moisture retention capacity (as found in regions of aridity) will have greater sensitivity and vulnerability to climate change. The Global Climate Observing System (GCOS) has defined soil moisture as an essential climate variable (ECV) which can be monitored at a regional and global scale from satellites measurements validated from a network of in-situ measurements (GCOS, 2015).
- Some researchers have found large emissions of rain-induced NO<sub>x</sub> (NO+NO<sub>2</sub>) from the soil following long periods of 20 drought in savannahs and seasonally dry forests (Jaegle et al., 2005;Feig et al., 2008;Hudman et al., 2010;Kim et al., 2012) and a strong link between this soil-released NO<sub>x</sub> and atmospheric NO<sub>2</sub> (Jaegle' et al., 2005;Delon et al., 2015). Microbial activities lead to denitrification and nitrification in the soil, resulting in the formation of soil gases such as NO (Delon et al., 2015). Kim et al. (2012) explained that root secretions from reviving plants following rewetting could significantly affect the soil surface flux of soil gases like NO. Water-stressed bacteria become active as soon as water drops on the dry soil
- 25 (Hudman et al., 2012) and feed on nutrients which have accumulated in dry season or longer periods between irregular/sporadic rainfalls (Meixner and Yang, 2006). NO is also produced rapidly after N (Nitrogen) fertilizers (both livestock manure and synthetic fertilizer) are applied to the soil to improve agricultural outputs (Pilegaard, 2013). The NO emissions from these microbial/bacterial activities and nitrification of the soil through fertilizers are released to the atmosphere through pulsing and oxidised in a reaction with  $O_3$  in the atmosphere to form NO<sub>2</sub> (Pilegaard, 2013) especially in
- 30 strongly sunlit areas of semi-aridity (Delon et al., 2008). It has been predicted that by the 2050s, soil emissions of  $NO_x$  are likely to decline by 9% in Northern (West and North) Africa while biomass burning emissions will be 12% more (Wai et al., 2014).

The West African region is situated between 4° N and 28° N latitude and 16° W and 15° E longitude, covering a total area of 6 million km<sup>2</sup> (one fifth of Africa) with the Gulf of Guinea as its southern boundary. The seasonal oscillation of the ITCZ (inter-tropical convergence zone) in the north-south direction over West Africa defines the climate zones (Conway, 2009) as it determines how much rainfall each climate zone gets annually. The Köppen-Geiger climate zones of West Africa (Kottek et al., 2006), which occur in latitudinal strata as shown in Figure 1 are:- the equatorial monsoon zone, equatorial winter zone, arid steppe zone and arid desert zone.

Figure 1: A map of West Africa showing the climate zones. The equatorial monsoon region is found along the West African boundary with the Gulf of Guinea. The arid steppe zone is often referred to as the Sahel.

- The annual rainfall over different zones is relatively constant but decreases from south to north away from the equator (Eltahir and Gong, 1996). However, drought and subsequent floods have been experienced in West Africa over the last few decades especially since 2002 (Tschakert et al., 2010), altering the annual rainfall cycles. Based on temperature and rainfall, there are two main seasons (dry and wet) in the humid areas of West Africa. These patterns vary for the semi-arid (arid steppe) / arid (arid desert) regions which have three main seasons:- a cool dry season between October and February, a warm
- dry season between March and June, and a warm wet season between July and September (Batello et al., 2004). In the other climate zones, there is one main dry season from November to February (the fire season is usually at its peak from December to February) and one main wet season: from June to August, with intermittent rainfall before and after the main wet season. FAO (1983) identifies two air masses which control rainfall and atmospheric transport around West Africa: the tropical

maritime (characterized by south-westerly winds coming to land off the Gulf of Guinea and bearing moisture) in the wet season and the tropical continental (originating from the Sahara desert bearing dryness) in the dry season.

In addition to climate variability, land/vegetation cover types may affect the soil-atmosphere exchange rates of soil  $NO_x$ 

- 5 (Feig et al., 2008). Figure 2 shows the percentage land cover of the West African climate zones aggregated from the ESA GLOBCOVER 2009 (Bontemps et al., 2010) land cover map. The equatorial climate zone was split into three subsets based on the location within West Africa (west, central and east equatorial monsoon). The equatorial winter region has the highest urban land cover spread across the zone and the West African coast including Abuja, Lagos, Accra and Dakar megacities. In addition to vast bare areas and sparse vegetation like in the arid desert zone (> 25 %), the arid steppe zone has the largest
- 10 percentage of regularly flooded vegetation (~ 4%) and rain-fed croplands (10 %) which are strongly affected by rainfall/soil moisture seasonality. The equatorial monsoon zones are covered by 25 30% broadleaved deciduous forests. The east equatorial monsoon with consists of the biodiverse wetlands of the Niger Delta has the highest percentage of wetlands (broadleaved flooded forest).