# Peer review of "Tropospheric NO2 concentrations over West Africa are influenced by climate zone and soil moisture variability"

_Atmospheric Chemistry and Physics, 2016_

## Referee Comment (RC1) · Anonymous Referee #1 · 31 Jan 2017

Review on manuscript acp-2016-1128

"Tropospheric NO2 concentrations over West Africa are influenced by climate zone and soil moisture variability " by A.R. Onojeghuo, H. Balzter and P. Monks

The objective of this article is to highlight the links between NO2 colums in the troposphere and soil water content, by using satellite data and statistical analysis, on different climate zones classified from arid to humid depending on their latitudinal position in West Africa. The topic is very interesting, and the tools used for the analysis give attractive figures. However the discussion lacks of in-depth interpretation and does not go far enough in the meaning of the results. I would recommend rejecting the paper at that stage of the work, but I would also recommend re-submitting it after careful proof

reading of the English language and thorough interpretation of the results which would give certainly very interesting conclusions. In that purpose, I give general and specific comments to improve the manuscript and consider a new submission.

General comments:

In the discussion part, the text remains very descriptive, and more interpretation of the results are needed. The conclusion does not reflect the title.

English language needs careful proof reading (sentences beginning by "with" should be written in another way).

Soil Water Index is deduced from satellite data, and is mentioned as soil moisture in the text, whereas figures mention SWI. Please choose one denomination only and homogenize throughout the text.

The land cover classes in Figure 2 are not exploited enough in the interpretation. Furthermore, the influence of anthropogenic emissions, biomass burning or natural emissions are discussed, but without any clear basis. Emission inventories would be needed to explain NO2 tropospheric columns, or at least information from literature.

The acronyms should be detailed the first time they are used.

Specific comments:

Abstract: mention the period used for this study. Line 22: emissions of NO2 do not exist. Should be emissions of NO or concentrations of NO2.

Introduction: Paragraph from lines 19 to 32 page 2 should be placed before paragraph from lines 6 to 17. Page 3, Figure 1: The Köppen Geiger climate classes mention a "tropical savanna" class. This class is never mentioned again in the text. Please use the same denominations to determine the climatic classes used in figure 1, in the text (especially page 4 lines 3 to 14) and in figure 2. Page 3 line 10: "relatively constant" is not precise enough. Please quantify and explain on which basis you give

this statement. Page 4 line 19: the sentence is too long and the same thing is repeated twice.

Methods: page 6 line 9: may be you could explain in a few words the role of each R package.

Page 7 line 16-17: "GC test. . . ". I do not understand this sentence.

Results and discussion page 8 This part should be reorganized and results could be discussed according to the vegetation types and the N emissions found in each climatic zone. Urban and traffic emissions are mentioned for Lagos et Abuja, what about flaring, which is mentioned later on in the manuscript?

Page 8 line 5: what are "soil moisture emissions"?

Page 9: Figure 4 needs a better analysis. What information does it bring compared to the other figures? The information seems to be given but should be better organized to help the reader follow the discussion. Paragraph from lines 17 to 21 is redundant with the preceding.

Page 10 line 20 "arid steppe zone" is repeated twice.

Page 11 line 10: sentence is not correct. Line 24: can you explain in more details why "the increasing soil moisture can cause an increase in the abundance of OH radical?" What about $NO_2$ deposition onto vegetation? It also a key loss process of $NO_2$, depending on the type of vegetation in presence. Line 32 to 35: the sentences are not correct.

Page 12: A lot of description of figure 6 is given but no reasons why $NO_2$ columns decrease when SWI increases in JJA and SON. $NO_2$ concentrations depend not only on NO emissions from soils or anthropogenic sources, but also on $NO_2$ deposition (and on the type of vegetation). The information is in the paper, but is not analyzed correctly.

Page 13 line 1: Feig et al. refers to Water Filled Pore Space, and not on soil moisture

or SWI. These are different ways of representing the soil water content and a direct comparison is not possible. Line 11, F should be 10.22 instead of 72.50 in the case of arid desert as mentioned in table 1. No interpretation is given in reference to this Granger causality test results, this is frustrating for the reader. Sentence line 15 is not correct.

Page 14 line 14: you mention "global effect of climate change induced soil moisture variability" but no explanation is given. What influences these flooding of the Niger Delta do have on NO2 columns?

Page 15 line 2: the Green wall initiative is mentioned: do you mean that this project has increased NO2 concentrations in the troposphere due to the input of fertilizers? In that case this has nothing to do with soil moisture variability? How do you cross this statement with previous results of figures 3, 4, 5, 6, 7? Same comment for NO2 decline and the decrease in gas flaring: what is the link with soil moisture?

Page 16: Conclusion. The text does not allow to conclude that "soil moisture plays a vital role in reducing atmospheric NO2". It is difficult to understand why.

---

## Referee Comment (RC2) · Anonymous Referee #2 · 19 Feb 2017

This paper is does not make a persuasive case that it has learned something new or that it has demonstrated any sort of useful link between soil moisture and NO2 emissions. I recommend the paper be rejected.

If the authors choose to revise I recommend a revised manuscript have 2-3 figures and no more. The figures should more directly address the authors claim of showing a causal and mechanistic relationship between soil moisture and NO2.

In addition, a revised manuscript should pay careful attention to time scales for rainfall and subsequent emissions, to separating seasonal cycles in transport and OH from other factors that affect NO2 columns, to removing the effects of biomass burning on NO2 columns, etc. Perhaps a convincing case about soil moisture could be made if

the paper started with a single climate zone and it illustrated how the soil moisture argument affects the NO2 column in a way that controls for these and other well known important variables. Another way to make a convincing case would be to show that the same methods of analysis applied to a 3-d model with and without soil NOx emissions produces meaningful differences.

In addition, a revised paper should carefully summarize current understanding of soil NOx emissions in the region so the reader has a clear understanding of what is new about the analysis and what aspects confirm prior results.

---

## Author Comment (AC1) · 20 Mar 2017

"This paper is does not make a persuasive case that it has learned something new or that it has demonstrated any sort of useful link between soil moisture and NO2 emissions. "

We have done a more extensive search of literature and can confirm that our findings are relevant to what is known of the relationship between soil moisture and NO2 over West Africa. Our discussion section has now been revised citing Zörner et al. (2016) to give credibility to the results presented in this paper.

"If the authors choose to revise I recommend a revised manuscript have 2-3 figures

and no more. The figures should more directly address the authors claim of showing a causal and mechanistic relationship between soil moisture and NO2."

We have revised the number of figures and merged Figures 3 and 4. We will consider merging a few more figures.

"In addition, a revised manuscript should pay careful attention to time scales for rainfall and subsequent emissions, to separating seasonal cycles in transport and OH from other factors that affect NO2 columns, to removing the effects of biomass burning on NO2 columns"

We indicated in the paper that biomass burning cycles play a vital role in the NO2 levels observed in the West African dry season. The dry season was determined relative to published literature and our observation from the annual soil moisture cycles. A reference to Roberts et al. (2009) has now been added to describe the months of active fires over the unique West African climate zones. For the timescales of emissions after rainfall, reference has been made to Zörner et al. (2016) who considered precipitation, water vapour, soil moisture and their impacts on emissions before and after rain events in the Sahel.

"Perhaps a convincing case about soil moisture could be made if the paper started with a single climate zone and it illustrated how the soil moisture argument affects the NO2 column in a way that controls for these and other well known important variables."

This is a good idea that could be explored further with future research themes.

"Another way to make a convincing case would be to show that the same methods of analysis applied to a 3-d model with and without soil NOx emissions produces meaningful differences."

This is also a very good idea but deviates from the theme of the research presented in this paper which was the use of existing open source satellite data to analyse the relationships between soil moisture and NO2.

"In addition, a revised paper should carefully summarize current understanding of soil NOx emissions in the region so the reader has a clear understanding of what is new about the analysis and what aspects confirm prior results." A section, which discussed soil emissions of NOx over West Africa, was included in the manuscript (Page 2 line 20 - ). Some literature consulted for this section include Hudman et al. (2012) and Jaegle et al. (2005) who have carried out NOx source partitioning globally. Zörner et al. (2016) has also carried out some valid research on soil NOx emissions from the West African Sahel and has been analysed in the reviewed manuscript. The "soil emissions of NOx" paragraph in the research has also been discussed and restructured as recommended by the first reviewer.

References

Hudman, R. C., Moore, N. E., Mebust, A. K., Martin, R. V., Russell, A. R., Valin, L. C., and Cohen, R. C.: Steps towards a mechanistic model of global soil nitric oxide emissions: implementation and space based-constraints, Atmos. Chem. Phys., 12, 7779-7795, 10.5194/acp-12-7779-2012, 2012. Jaegle, L., Steinberger, L., Martin, R. V., and Chance, K.: Global partitioning of NOx sources using satellite observations: Relative roles of fossil fuel combustion, biomass burning and soil emissions, Faraday Discuss, 130, 407-423, 2005. Roberts, G., Wooster, M. J., and Lagoudakis, E.: Annual and diurnal african biomass burning temporal dynamics, Biogeosciences, 6, 849-866, 2009. Zörner, J., Penning de Vries, M., Beirle, S., Sihler, H., Veres, P. R., Williams, J., and Wagner, T.: Multi-satellite sensor study on precipitation-induced emission pulses of NOx from soils in semi-arid ecosystems, Atmos. Chem. Phys., 16, 9457-9487, 10.5194/acp-16-9457-2016, 2016.

---

## Author Comment (AC2) · 20 Mar 2017

"The topic is very interesting, and the tools used for the analysis give attractive figures. However the discussion lacks of in-depth interpretation and does not go far enough in the meaning of the results. I would recommend rejecting the paper at that stage of the work, but I would also recommend re-submitting it after careful proof reading of the English language and thorough interpretation of the results which would give certainly very interesting conclusions."

Thank you for your valid observation. The results have now been discussed further with reference to relevant literature such as Zörner et al. (2016). Some more proof reading is also being done.

"In the discussion part, the text remains very descriptive, and more interpretation of the results are needed. The conclusion does not reflect the title. English language needs careful proof reading (sentences beginning by "with" should be written in another way)."

The conclusion section is being re-written to reflect the core results of this research.

"Soil Water Index is deduced from satellite data, and is mentioned as soil moisture in the text, whereas figures mention SWI. Please choose one denomination only and homogenize throughout the text."

This has now being homogenised with a sentence explaining that we have chosen the SWI an indicator for soil moisture levels at the surface.

"The land cover classes in Figure 2 are not exploited enough in the interpretation. Furthermore, the influence of anthropogenic emissions, biomass burning or natural emissions are discussed, but without any clear basis. Emission inventories would be needed to explain NO2 tropospheric columns, or at least information from literature."

The influence of anthropogenic emissions is discussed as beyond soil moisture, they also play a role in NO2 variability over West Africa at certain times of the year. More references to literature including Jaegle et al. (2005) who carried out a global partitioning of NOx emission sources and Roberts et al. (2009) have been inserted into the manuscript.

"The acronyms should be detailed the first time they are used. Abstract: mention the period used for this study. Line 22: emissions of NO2 do not exist. Should be emissions of NO or concentrations of NO2. Introduction: Paragraph from lines 19 to 32 page 2 should be placed before paragraph from lines 6 to 17. "

These sections have been reformatted as suggested. Line 22 has been rephrased to reflect emissions of NO rather than NO2.

"Page 3, Figure 1: The Köppen Geiger climate classes mention a "tropical savanna" class. This class is never mentioned again in the text. Please use the same denomi-
nations to determine the climatic classes used in figure 1, in the text (especially page 4 lines 3 to 14) and in figure 2. "

The "tropical savanna" in the Koppen Geiger climate classes was a labelling error and has now been corrected. Attached is a revised image of the study area to put other parts of the manuscript in proper context.

"Page 3 line 10: "relatively constant" is not precise enough. Please quantify and explain on which basis you give this statement. Page 4 line 19: the sentence is too long and the same thing is repeated twice."

The repeated portion has been deleted. Page 3 line 10 has been further discussed. Page 3 line 10 had to do with the minimal variations in the amount of rainfall over each climate zone from year to year. This has been re-written. "Methods: page 6 line 9: may be you could explain in a few words the role of each R package."

The role of each R package has been indicated with a few words in the manuscript.

"Page 7 line 16-17: "GC test: : : ". I do not understand this sentence."

Granger Causality has been written in full here to ensure the message is passed across as intended. The sentence was to explain that the Granger causality tests are performed to see if past values of a variable can add to the explanatory autoregressive model of that variable and another.

"Results and discussion page 8 This part should be reorganized and results could be discussed according to the vegetation types and the N emissions found in each climatic zone."

We have retained the discussions according to climate zones with improved reference to the type of GLOBCOVER landcover class/vegetation found in these unique climate zones

"Urban and traffic emissions are mentioned for Lagos et Abuja, what about flaring,

**ACPD**
which is mentioned later on in the manuscript?"

A referenced sentence has been added in the introduction section to indicate that gas flaring in West Africa is most prevalent in the Niger Delta which is located in the east equatorial monsoon zone.

"Page 8 line 5: what are "soil moisture emissions"?"

This was an error and has now been corrected to "Areas in the arid steppe climate zone where soil moisture variations may induce soil NO emissions had tropospheric NO2 concentrations

decrease when SWI increases in JJA and SON. NO2 concentrations depend not only on NO emissions from soils or anthropogenic sources, but also on NO2 deposition (and on the type of vegetation). The information is in the paper, but is not analyzed correctly."

This will be discussed better in the revised manuscript as recommended.

"Page 13 line 1: Feig et al. refers to Water Filled Pore Space, and not on soil moisture or SWI. These are different ways of representing the soil water content and a direct comparison is not possible."

This portion of the manuscript has been rephrased to indicate that Feig et al. used the Water Filled pore space as an indication for soil moisture. We have used SWI instead and found our results to be similar. "Line 11, F should be 10.22 instead of 72.50 in the case of arid desert as mentioned in table 1. No interpretation is given in reference to this Granger causality test results, this is frustrating for the reader. Sentence line 15 is not correct."

We apologise as the F value stated was an error and has been corrected. The results have now been interpreted better to indicate that F values for the arid steppe and arid desert climate zones validate what is already known about the impact of soil moisture on NOx variations.

"Page 14 line 14: you mention "global effect of climate change induced soil moisture variability" but no explanation is given. What influences these flooding of the Niger Delta do have on NO2 columns?"

This is being looked at in closer detail relative to published literature on the impact of soil moisture trends on NO2 variability in humid regions.

"Page 15 line 2: the Green wall initiative is mentioned: do you mean that this project has increased NO2 concentrations in the troposphere due to the input of fertilizers? In that case this has nothing to do with soil moisture variability? How do you cross this

**ACPD**
statement with previous results of figures 3, 4, 5, 6, 7? Same comment for NO2 decline and the decrease in gas flaring: what is the link with soil moisture?"

The Green wall initiative was mentioned as NPK fertilizers, which are known to affect NO emissions especially in arid soils is being used to improve vegetation re-growth in the Sahel. Slemr and Seiler (1984) showed that NPK fertilizers had a stronger effect of soil NO emissions than other fertilizer types. This portion of the manuscript will be revised.

Page 16: Conclusion. The text does not allow to conclude that "soil moisture plays a vital role in reducing atmospheric NO2". It is difficult to understand why. The conclusion of the research will be discussed better in the revised manuscript.

**Reference**

Jaegle, L., Steinberger, L., Martin, R. V., and Chance, K.: Global partitioning of NOx sources using satellite observations: Relative roles of fossil fuel combustion, biomass burning and soil emissions, Faraday Discuss, 130, 407-423, 2005. Roberts, G., Wooster, M. J., and Lagoudakis, E.: Annual and diurnal african biomass burning temporal dynamics, Biogeosciences, 6, 849-866, 2009. Slemr, F., and Seiler, W.: Field measurements of NO and NO2 emissions from fertilized and unfertilized soils, Journal of Atmospheric Chemistry, 2, 1-24, 1984. Zörner, J., Penning de Vries, M., Beirle, S., Sihler, H., Veres, P. R., Williams, J., and Wagner, T.: Multi-satellite sensor study on precipitation-induced emission pulses of NOx from soils in semi-arid ecosystems, Atmos. Chem. Phys., 16, 9457-9487, 10.5194/acp-16-9457-2016, 2016.

**ACPD**
Fig. 1. Revised map of West Africa showing the Koppen Geiger classes

**ACPD**